# Unveiling CNS cell morphology with deep learning: A gateway to anti-inflammatory compound screening

**Hyunseok Bahng, Jung-Pyo Oh, Sungjin Lee, Jaehong Yu, Jongju Bae, Eun Jung Kim**
**Sang-Hun Bae** \*, Ji-Hyun Lee\*

Research Center, DR. NOAH BIOTECH Inc., 91, Changnyong-daero 256beon-gil, Yeongtong-gu, Suwon-si, Gyeonggi-do , Republic of Korea

\* shbae@drnoahbiotech.com (SH), jhlee@drnoahbiotech.com (JH)

## Abstract

Deciphering the complex relationships between cellular morphology and phenotypic manifestations is crucial for understanding cell behavior, particularly in the context of neuropathological states. Despite its importance, the application of advanced image analysis methodologies to central nervous system (CNS) cells, including neuronal and glial cells, has been limited. Furthermore, cutting-edge techniques in the field of cell image analysis, such as deep learning (DL), still face challenges, including the requirement for large amounts of labeled data, difficulty in detecting subtle cellular changes, and the presence of batch effects. Our study addresses these shortcomings in the context of neuroinflammation. Using our in-house data and a DL-based approach, we have effectively analyzed the morphological phenotypes of neuronal and microglial cells, both in pathological conditions and following pharmaceutical interventions. This innovative method enhances our understanding of neuroinflammation and streamlines the process for screening potential therapeutic compounds, bridging a gap in neuropathological research and pharmaceutical development.

## Introduction

Understanding cell behavior necessitates deciphering the intricate relationships between cellular morphology and the phenotypic manifestations cells exhibit in response to internal and external stimuli [1,2]. For instance, a recent study has underscored that the distinct morphological features of cancer cells accurately delineate their tumorigenicity and metastatic potentials [3]. The morphological changes of microglia, the primary immune cells in the CNS, are known to be highly correlated with the activation states of the cells [4]. It has also been shown that small molecule perturbations can cause cellular changes as clearly observed in morphological differences [5–7], rendering cell-based screening routine in compound discovery. Despite the substantial progress in morphology studies, cellular image-based phenotypic screening has received less attention than specific biological assays during the initial step of compound discovery [3,8–11]. Conventionally,

**Data availability statement:** The datasets used and analyzed during the current research are available from the following URL: https://doi.org/10.5281/zenodo.10369052.

**Funding:** The author(s) received no specific funding for this work.

**Competing interests:** The authors have declared that no competing interests exist.

morphology analysis has been considered a labor-intensive procedure involving microscopic imaging, object segmentation, and feature extraction to produce quantitative data. In addition, accurate identification of cellular changes requires expert curation, and subtle cellular changes in response to varying degrees of external or internal stimuli are hard to discern even for experts. These limitations hampered using image-based analysis for screening large amounts of compounds.

Recent advancements in automated microscopes and the development of state-of-the-art image analysis algorithms based on DL have diminished hitherto labor-intensive procedures in cellular morphology analysis [12]. With the success of convolutional neural networks (CNNs) in the field of image processing, the DL-based image analysis has outperformed conventional approaches in terms of accuracy and robustness [13,14]. The capability to detect hierarchical structures and identify complex patterns in image data [15] has facilitated the broad usage of CNNs in the field of visual tasks. Recently, a vision Transformer (ViT), an attention-based algorithm, has surfaced as a novel approach [16], complementing the established CNNs. The DL-based algorithms are increasingly employed for common tasks in biomedical images, particularly in classification [17], digital pathology [18] and cell or nuclear segmentation [19]. Furthermore, the democratization of access to these sophisticated algorithms has been facilitated by open-source frameworks such as PyTorch [20] and tensorflow [21]. Consequently, cellular image-based phenotypic screening can concentrate more on addressing the biological questions at hand than the technical difficulties.

Meanwhile, the sensitive nature of biological data makes it susceptible to non-biological technical factors such as acquisition protocols, temperature variations, and ozone levels [22,23]. Even with meticulous experimental controls, distinct patterns may emerge across different batches, a phenomenon commonly known as batch effect. In this study, experiments were conducted to obtain image data from cells cultivated on each respective plate, revealing the presence of batch effects among these plates. To address this issue, an ensemble-based decision-making method was employed.

Primary brain cell culture, commonly used for CNS targeting compound development, is an appropriate system for taking advantage of DL-based image analysis. Culture contains heterogenic cell types, including neurons, astrocytes, and microglia, which should be precisely isolated to measure cell type-specific changes. Furthermore, their intrinsic shape makes it difficult to identify the boundaries of cells and segmentation from microscopic images. Stimulants mimicking pathological conditions induce morphological changes in brain cells, which can be effectively detected by DL-based image analysis.

The aim of this study was to develop a phenotypic screening system to evaluate the anti-inflammatory effect of test compounds by DL-based image analysis. Initially, due to the limited amount of research on the structure and conditions of CNS cells, we employed advanced deep learning techniques to establish a reliable connection between the cellular morphologies of neuronal and microglial cells and their corresponding neuropathological conditions. This approach allows us to delve into the intricate relationship between the subtle alterations in cellular structures and their phenotypic irregularities. To mitigate the batch effects in our data, we utilized an ensemble-based strategy to enhance the reliability of the prediction. Lastly, we confirmed the biological relevance of the DL-based screening system by evaluating previously known anti-inflammatory compounds. This pipeline efficiently classified the varying degree of neuroinflammation based on dynamic cellular morphological changes and evaluated the effects of potential pharmaceutical interventions on the cellular states.

## Result

### Deep learning-enhanced phenotypic screening of anti-inflammatory compounds in mouse primary brain culture

Our research employed a DL-based phenotypic screening of compounds that are effective against neuroinflammation. In the first step, we generated ground truth data for inflammation severity by producing thousands of cellular images at different levels of inflammatory state. Cortical cells were isolated from mouse embryonic brain and seeded onto culture plates. To generate training images for the DL model, cells were exposed to Lipopolysaccharides (LPS) at different concentrations to induce various inflammatory states (0.005–20 μg/ml). Cells were immunostained with antibodies specific for neuronal and microglial markers and visualized under fluorescent microscopy. To annotate training images in different inflammatory states, ground-truth labels in the pipeline, we classified the microscopic images based on the treated concentration of LPS. This labeling process, along with the well-established experiment, eliminated the need for manually curating the cellular states. This approach also allowed the separation of subtle morphological changes that would otherwise be indistinguishable to human eyes and facilitated the classification of such variations into distinct severity categories (Fig 1A).

To improve the image quality and ensure the generalization of the DL model, the cell images were preprocessed into a 3-channel format and underwent a series of image augmentations. The preprocessed images, along with their corresponding severity levels, were fed into a state-of-the-art model for classification training. In this study, we extensively explored various CNNs and ViT to discern their respective strengths and capabilities. Our aim was to identify the optimal model for effectively classifying the cell images within our dataset. Subsequently, we introduced a novel ensemble-based approach to predict results, which proved to be highly effective in mitigating batch effects. This method addressed the challenge posed by batch effects that can emerge during experiments, particularly when dealing with multiple plates or batches of data. Finally, we conducted a rigorous assessment of compound efficacy using the model (Fig 1B).

### Transforming cellular microscopic images for deep learning: staining, augmentation, and normalization techniques

In this study, we utilized microscopic images labeled with Nuclei, Neuron, or Microglia cells. These images have been specifically stained to enhance visualization under the naked eye. While these images are suitable for direct observation, it is essential to preprocess them into a format more conducive to deep learning algorithms.

The dataset used in this study was obtained by staining neuronal cells with MAP2(Green), microglia cells with Iba-1(Red), and nucleic acids with DAPI(Blue). This staining process specifically highlighted the target cells, enabling the collection of high-quality images for analysis. (Fig 2A) Following staining, the cell images in a 3-channel format were cropped into 15 overlapping segments (Fig 2B), and before initiating the learning process, we performed a channel-specific preprocessing of the 3-channel (RGB) image data. Specifically, we separated the image into individual channels, assigning the red channel to Iba-1, the green channel to MAP2, and the blue channel to Nucleus. This preprocessing step ensures that our model learns from distinctly classified channels, each representing different biological structures.

Due to the diverse characteristics of the cellular images, preprocessing was conducted prior to model training to ensure consistent input for the deep learning algorithms. Preprocessing tasks were categorized into three main types: data augmentation, data conversion, and data

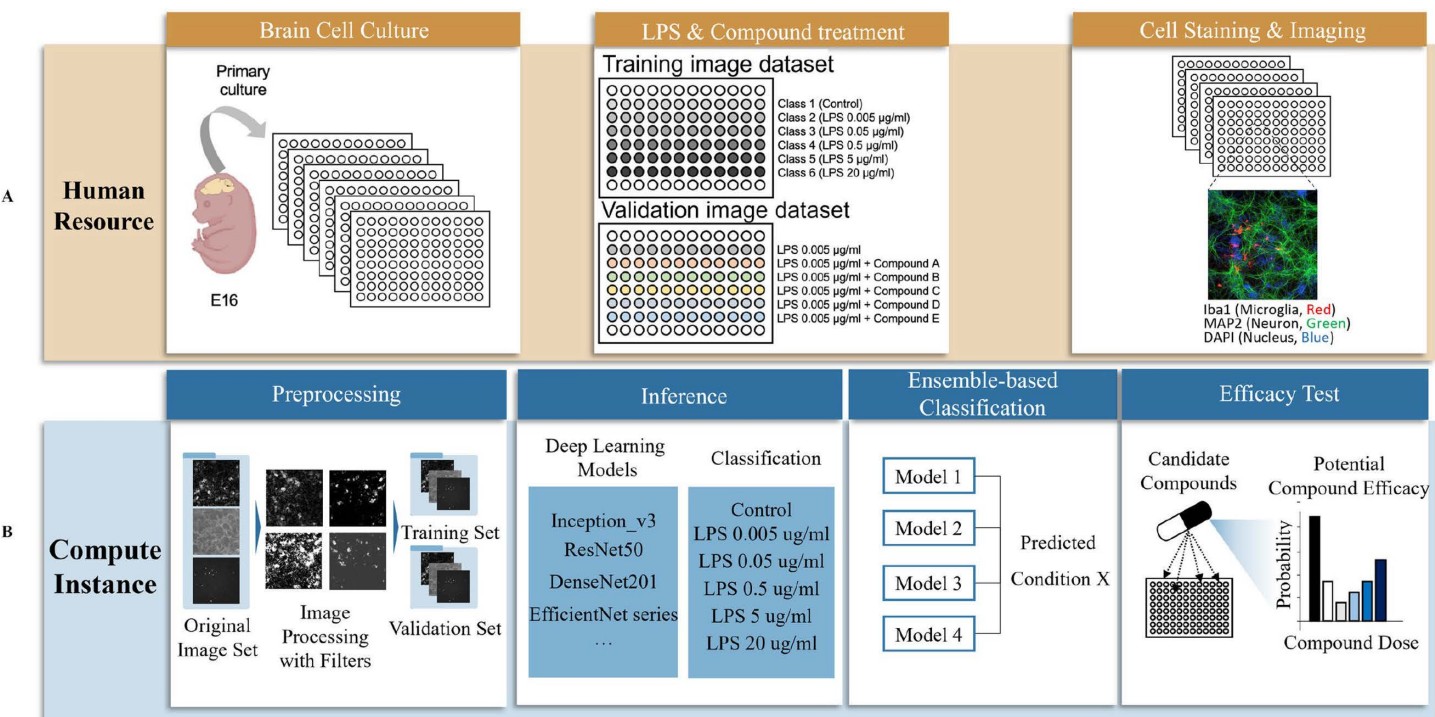

**Fig 1. An overview of the image generation and deep learning for phenotypic screening of compounds.** This approach consists of two distinct steps. (A) The cultivation of brain cells was followed by staining the cells under normal and pathological conditions. The stained cells were imaged with a fluorescence microscope. (B) The cell images were preprocessed using a variety of techniques to ensure they are appropriately optimized for deep learning. Our system incorporated an advanced classification based on deep learning, complemented by an ensemble method for quantitative estimation of cellular states. Ultimately, it was verified through efficacy testing of anti-inflammatory compounds.

normalization. In data augmentation [24,25], various techniques were employed to prevent overfitting and increase data diversity. These techniques involved applying various transformations to the images, such as shift, rotation, flip, enlargement, and reduction of image coordinates. To increase dataset size and variability, we augmented approximately 10% of the dataset by applying various transformations: shift, rotation, flip, enlargement, and reduction of image coordinates (Fig 2C, 2D). This level of augmentation was selected to balance enhancing the model's robustness and preserving the integrity of the original data. While augmentation is beneficial for mitigating overfitting, excessive augmentations may introduce noises that diverge from biological characteristics of the original data, potentially impact model performance. By limiting augmentation to 10%, the data was diversified while minimizing the risk of over-generalization (Fig 2C, 2D). Data conversion techniques were applied to improve model performance by modifying the characteristics of the data. These techniques included solarization, Multiplicative Noise, and RGB Shift. Data normalization was performed to standardize the intensity values of the images. This ensured that all images had a similar range of values, which improved the performance of the deep learning algorithms.

## Comparative analysis of CNNs and ViT architectures for classification: a comprehensive benchmark study

To facilitate the utilization of deep learning models for classification, we rigorously assessed the performance of two advanced architectures, CNNs and ViT. The two primary algorithms, CNNs and ViT, employ distinct methodologies to analyze images: CNNs automatically learn

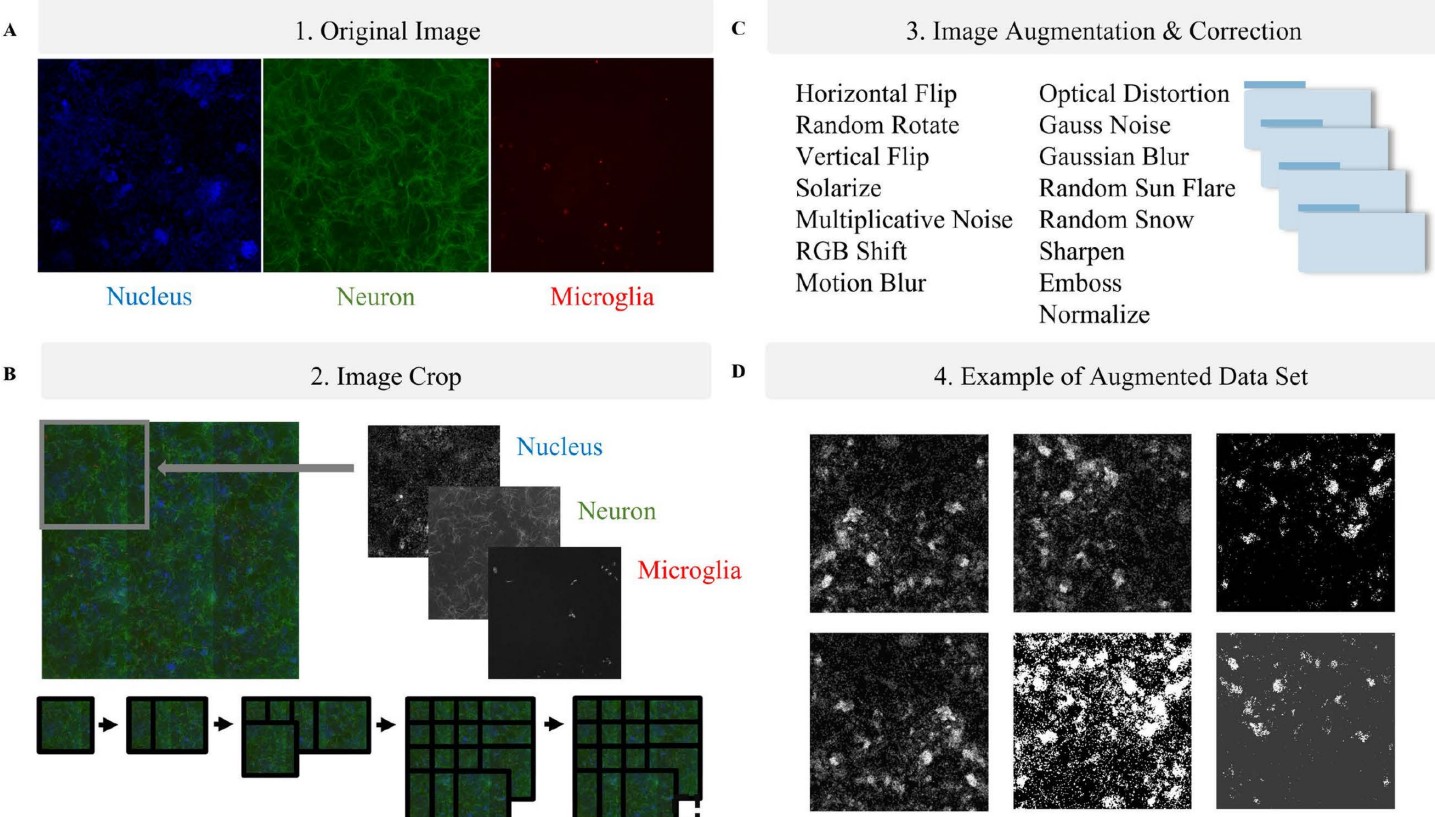

**Fig 2. Enhancing Image Quality with openCV Filters and Mathematical Techniques for Efficient Deep Learning Model Training.** (A) The cultured brain cells were immunostained with three markers: Iba-1 (Microglia) shown in red, MAP2 (Neuron) in green, DAPI (Nucleus) in blue. (B) Raw images of cells were cropped into 15 patches, allowing for overlap between adjacent cropped images to ensure comprehensive coverage. (C) The image dataset was enhanced and refined using a range of image augmentation and correction methods. (D) Representative examples of cell images after augmentation.

different layers of details in local regions of an image (e.g., edges, textures, and shapes) in the data (Fig 3A) [26,27], whereas ViT exhibit proficiency in capturing long-range dependencies across the entire image, enabling them to understand the relationship between different parts of the image (Fig 3B) [15,28]. To capitalize on the distinctive benefits of each, we utilized 11 CNNs and 3 ViT models. The comprehensive benchmark test allows us to develop a more precise and resilient pipeline for enhanced analysis of complex cell images.

In our findings, we included the published benchmark ImageNet results of all the models for comparison. Remarkably, the results, as shown in Fig 3C, revealed a significant difference in the F1 score of classifying cell states between two leading architectures. CNNs significantly outperformed ViT in classifying cell states, highlighting a clear difference between the two leading architectures.

EfficientNet-B5 achieved the highest F1 score (88%) at the well level (Fig 3C), where each well represents an individual experimental unit within a plate. We obtained this performance using a modified Leave-One-Out Cross-Validation (LOOCV) [29] approach with four plate groups, each serving as a validation set. The outcome is especially interesting, considering ViT's large number of parameters yielded high accuracy on the ImageNet dataset but performed poorly on our cell dataset. Our analysis showed the robust performance of CNN models in cell image analysis and emphasizes the need for careful determination of model architecture, considering the specific characteristics of the dataset at hand. Further details on the modified LOOCV approach are provided in the next section.

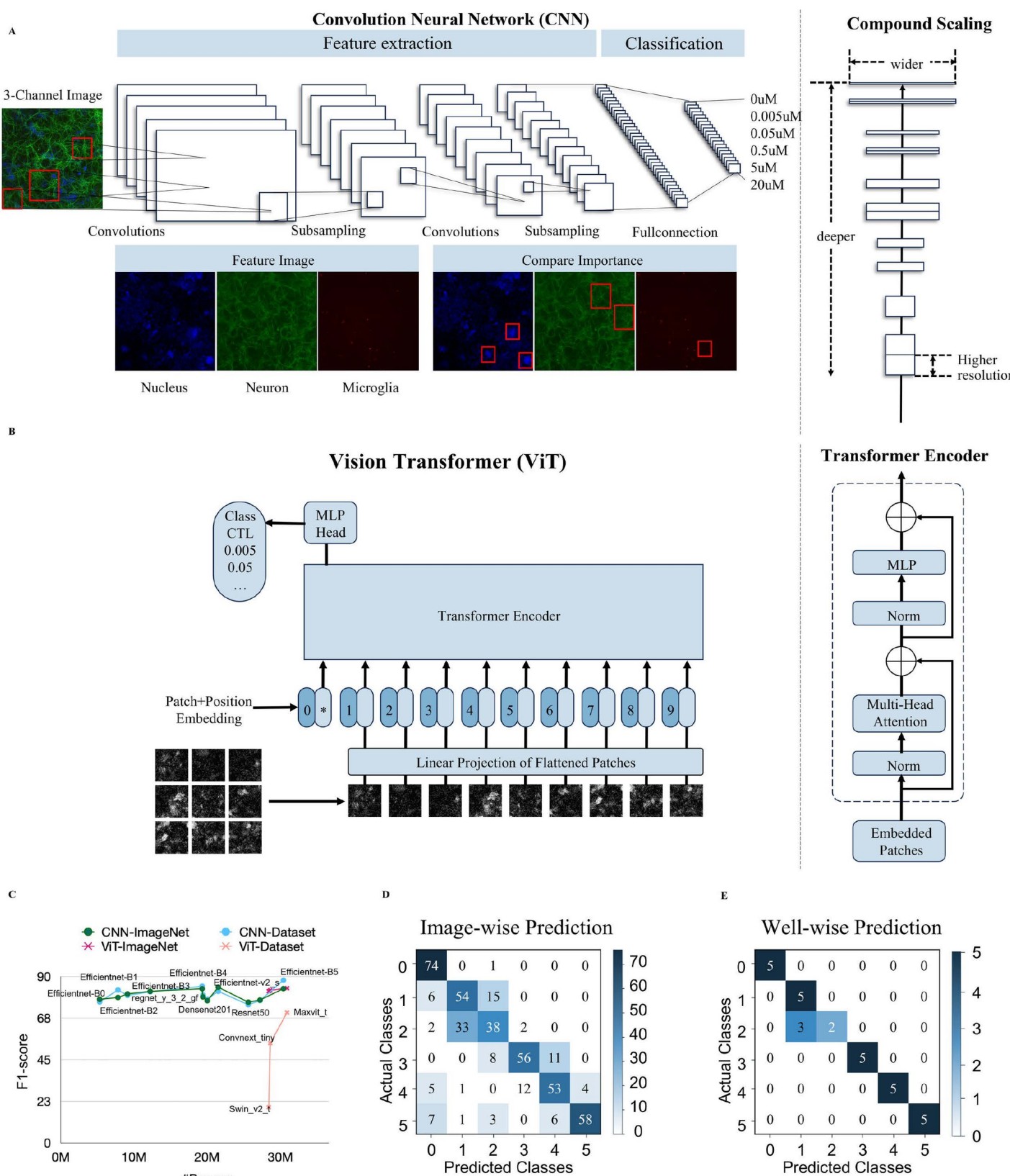

**Fig 3. Using two advanced deep learning architectures for optimal model selection.** (A) General CNN models architecture illustrating the standard layers and flow of cell image data on the left side. Efficient model architecture depicting three scaling factors (depth, width, and resolution) on the right side. (B) Overview of

Vision Transformer model architecture on the left side and a detailed description of the transformer encoder on the right side. (C) Comparison of F1 score across 14 different models. (D) Confusion matrix of the best model presents the number of cell images classified into each category. (E) Confusion matrix of the model shows the number of wells classified into each category. Label 0 in the y-axis represents the control group and the other groups are numerically labeled in accordance with the increasing concentration of LPS (the severity of neuroinflammation): 0 = control, 1 = 0.005 μg/ml, 2 = 0.05 μg/ml, 3 = 0.5 μg/ml, 4 = 5 μg/ml, 5 = 20 μg/ml.

The findings from the performance evaluation were examined in the context of a multiclass task, with an emphasis on the selected model's ability to classify varying degrees of neuroinflammation. To further evaluate the model, we used two approaches to define an entity that the learned model predicts: a cell image (image-wise predictions, Fig 3D) and a well of 15 cell images (well-wise predictions, Fig 3E). By considering each image as an instance, we could capture the variability within a single well, identifying cellular status across diverse cellular status on a granular level. In contrast, the rationale behind selecting a well as a separate entity reflects accurate classification as a macro level. At both levels, the best model accurately categorized the normal group against all pathological classes, as shown in the confusion matrix (Fig 3D, 3E). This high level of F1 score was also demonstrated by its capacity to identify the most severe pathological condition, marked by the highest concentration of LPS treatment. The model exhibited a slightly lower F1 score in classifying intermediate levels in the image-wise predictions, but it showed an impressive F1 score across all pathological states in the well-wise predictions. The result of dual-perspective analysis demonstrates the selected model's versatility and robustness to discern cellular states effectively.

## Ensemble-based batch effect mitigation

To mitigate confounding plate effects, we modified the data split strategy in our model development step (Fig 4A) and adopted an ensemble-based approach (Fig 4B) for predicting the efficacy of anti-inflammatory compounds.

In the model development process, we utilized leave-plates-out cross-validation, a modified version of LOOCV (Fig 4A) to better address plate variability and minimize overfitting. The standard LOOCV, which sequentially selects individual data points like single cropped images for validation, is not suitable for plate-based assay datasets. This approach often inflates performance metrics because predictive models can more easily learn patterns specific to the plates they were trained on, compromising their generalization ability. To address this limitation, we strategically assigned all wells from a single plate to the same group for validation, instead of using individual images. In our dataset, the seven plates were systematically combined into four groups. Three groups were formed by combining two plates, where one group was left with a single plate. During cross-validation, one group served as the validation set while the remaining groups formed the training set. This approach ensured that cell images from the same plate remained together in the same group, preventing data leakage. The validation set contained either 900 images when using groups 1-3 (2 plates × 450 images per plate) or 450 images when using group 4 (1 plate × 450 images) (Fig 4A).

In the evaluation step, predictions were generated using the four models developed in the preceding step, leading to four distinct predictions per image. As illustrated in Fig 4B, the final well prediction was calculated by rounding the average of the 60 results obtained from the 15 cropped images, with 4 predictions obtained for the multiple models. Formally, the final well prediction can be expressed as shown in Equation (1):

$$Round\left(\sum_{15}^{i=1}\sum_{4}^{j=1}\frac{R_{i,j}}{15\times4}\right) \tag{1}$$

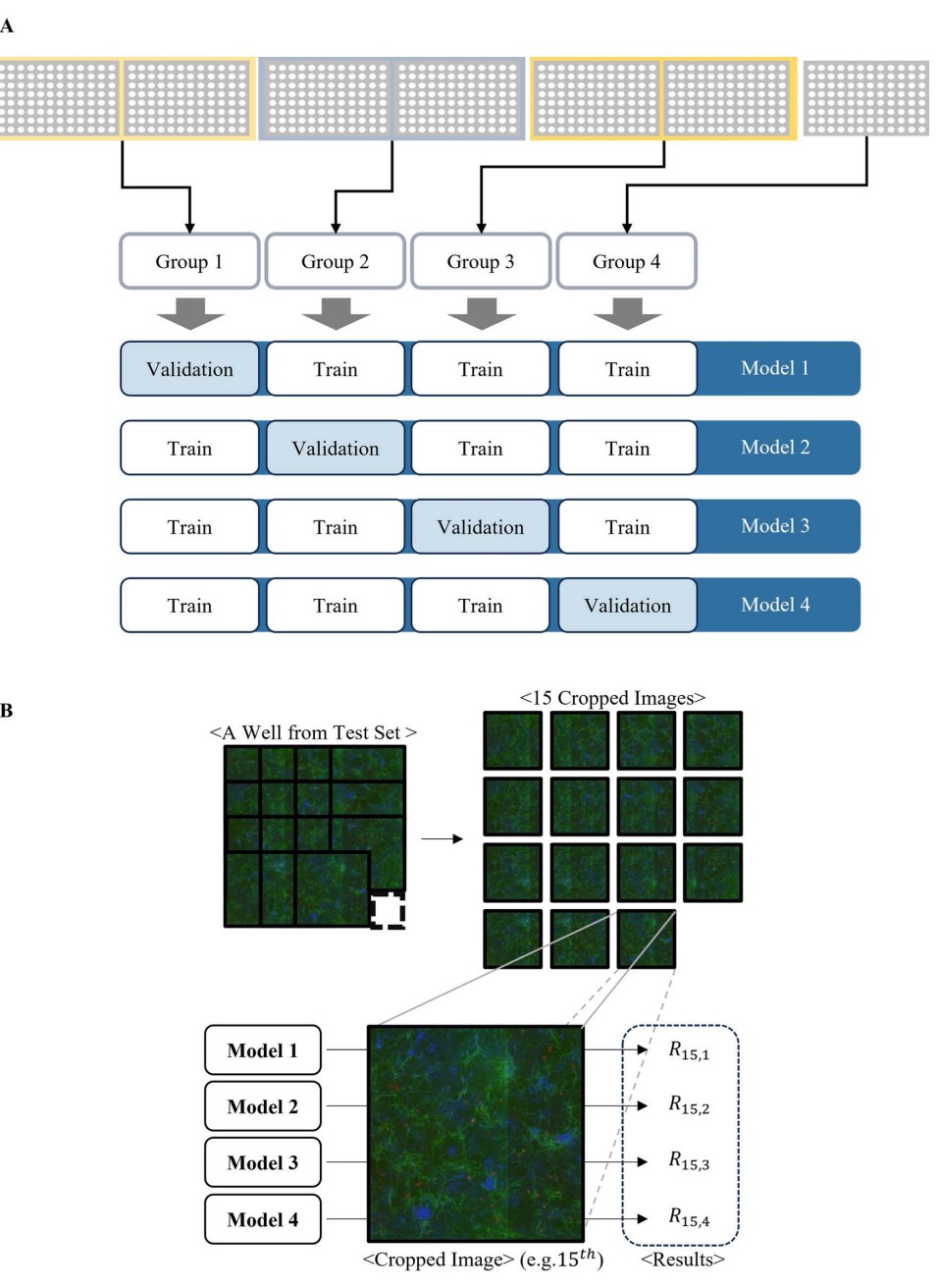

**Fig 4. Ensemble-based image classification.** (A) Model development step: Model development set, for instance, derived from seven distinct plates, is partitioned into four groups, each corresponding to a plate set. All images from the sample plates are assignedinto identical groups instead of being randomly distributed across the all groups(leave-plates-out cross-validation). This process results in the fitting of four distinct models through a sequence of four trials. (B) Evaluation step: In a single well, 15 cropped images are produced and evaluated using the four previously established models. For example, each of the four models produces distinct outcomes for the 15th cropped image.

The variable $R_{i,j}$ is an integer, taking on values of either 0 or 1, which signifies the category predicted by model j for the $i^{th}$ cropped image of the well. A value of 1 indicates that the model predicts the cropped image as a control (CTL), while 0 indicates LPS-treated. When the calculated value, rounded to the nearest integer, equals 1, the corresponding well image is predicted

as CTL. Here, LPS-treated refers to samples exposed to a mild concentration (0.005 μg/ml) rather than higher concentrations. This mild concentration induces reversible inflammation that can be modulated by therapeutic compounds, facilitating the meaningful assessment of anti-inflammatory treatments.

## Statistical analysis for model comparison

To assess the effectiveness of the ensemble approach, we compared the performance of the ensemble-based model against the best model from cross-validation. All the predictions were performed on images of the wells in the test set. Specifically, the test set included 6 wells (90 images) for CTL and 6 wells (90 images) for LPS-treated conditions, resulting in a total of 180 images. This comparison highlights the potential advantages of ensemble modeling in improving classification performance. Specifically, we evaluated the models' ability to predict outcomes for Control and LPS-treated images, focusing on key metrics such as the F1 score. The best model and the ensemble-based model achieved the F1 scores of 69% and 77%, respectively, and to assess the significance of the observed differences between the two models, McNemar's test [30] was conducted.

McNemar's test assesses whether the counts of images misclassified by one model but correctly classified by the other are statistically different. The test statistic is calculated as $(a-b)^2/(a+b)$ which follows a $\chi^2$ distribution with 1 degree of freedom. Here, the variable $a$ represents the number of images misclassified by the best model but correctly classified by the ensemble-based model, while the variable $b$ represents the number of images correctly classified by the best model but misclassified by the ensemble-based model. The ensemble-based model correctly classified 32 samples that the best model misclassified, while the best model correctly classified only 1 sample that the ensemble-based model misclassified (Fig 5A). This significant performance difference (Fig 5B) was confirmed by McNemar's test ($\chi^2 = 27.27$, p value < 0.001). These results suggest that the ensemble-based model outperforms the best model from the cross-validation in the image classification task.

## Dose-dependent evaluation of representative anti-inflammatory compounds through our phenotype screening pipeline

While our model was trained to classify the varying degrees of pathological states (Fig 3C, 3D), its ultimate aim is to facilitate practical application in the drug discovery process. To meet this final goal, instead of categorizing the cellular status in a discrete manner, we have shifted our focus towards quantitatively measuring the status by analyzing the ratios of the images predicted as control. Such an approach ensures that the model is not only accurate in classifying the pathological states, but also useful in real-world drug discovery scenarios.

Accordingly, we evaluated whether the model accurately captures the dose-dependent efficacy of known anti-inflammatory compounds, Resatorvid, Dexamethasone, and Dimethyl Fumarate (DMF). The evaluation hypothesized that the anti-inflammatory compounds might restore the pathological states of cells primed with LPS and increase the number of cells predicted as controls in a dose-dependent manner. Our model is expected to accurately detect these patterns in cell images of three markers combined. In addition, we investigated the individual contributions of each marker in accurately classifying the efficacy of compounds. This involved training separate models on single channel cell images and then conducting a comparative analysis. For assessment, cells treated with LPS at 0.005 μg/ml were subjected to increasing concentrations of the compounds and classified by our model.

The outcomes of the evaluation process were quantitatively assessed by determining the proportions of cropped cell images our model classified as controls. This metric effectively

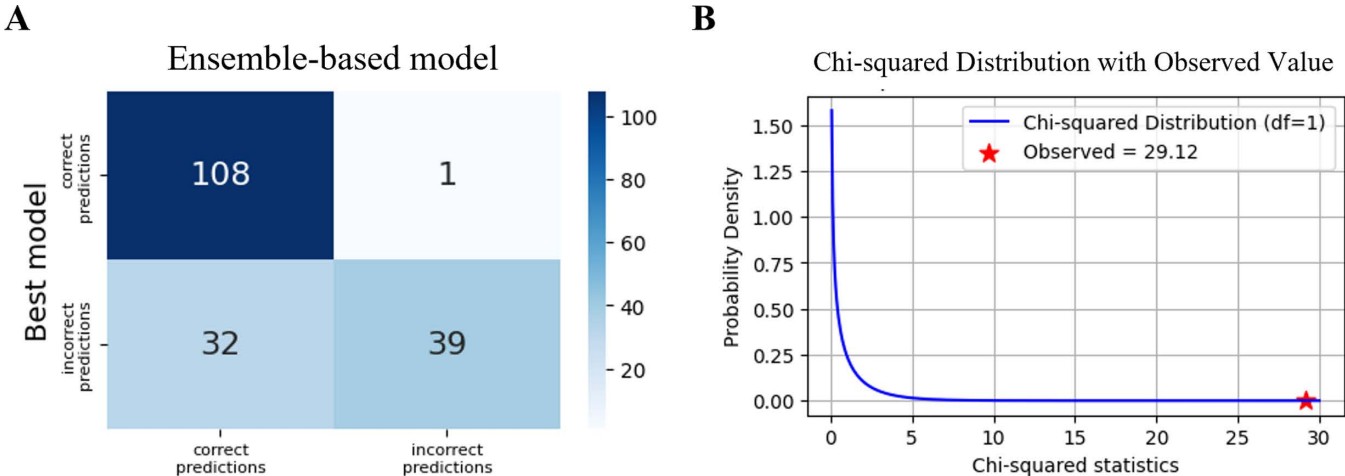

**Fig 5. Results of McNemar's test.** (A) Contingency table for classification agreements and disagreements between the best model and the ensemble-based model: upper-left (both correct), upper-right (ensemble incorrect, best correct), lower-left (ensemble correct, best incorrect), lower-right (both incorrect). (B) $\chi^2$ distribution with 1 degree of freedom showing significant difference between the two models by McNemar's test (observed $\chi^2$ statistic = 29.12, p value < 0.001). The observed test statistic is indicated by a red symbol (*).

measures the extent to which the medications induce the inflamed cells to return to their normal states. In three channel experiments with all three compounds, we observed a consistent, dose-dependent increase in the number of cell images classified as controls (Fig 6A-C), suggesting a strong correlation between compound dosage and cellular response. Across each channel, the Iba-1 marker was the only one to demonstrate a pattern that aligned with those observed in all three channels. This was supported by the fact that there was a highest correlation coefficient, exceeding 0.7, between the predictions from the Iba-1 makers and those from the combined markers (Fig 6D). These findings strongly reinforce the well-documented link between microglia cells and neuroinflammation, highlighting the crucial role of these cells in a range of neuropathological disorders.

We evaluated the efficacy of medications with established mechanisms of action and obtained results that support the initial hypothesis, affirming the applicability of our technique in the compound efficacy system. Notably, our pipeline proved effective at analyzing microglia cells and measuring neuroinflammation. These findings underscore the potential of our approach to efficiently uncovering compounds for addressing inflammation.

## Discussion

The principal strength of phenotypic screening lies in its ability to identify potential compounds without requiring prior knowledge of specific molecular targets. This target-agnostic approach proves especially valuable in discovering compounds for complex pathologies such as neurological diseases. Advancements in cell-based phenotypic screening tools have further enhanced the potential of this approach [31]. Based on these advancements, we have developed a DL-based phenotypic screening system to assess the anti-inflammatory effects (S1 Fig) in the primary cultured cells, incorporating an ensemble strategy to minimize batch effects.

To determine the optimal model for our task, we conducted a thorough comparison between the CNN and ViT models, investigating their distinct capabilities [27]. CNNs, renowned for their adeptness in hierarchical feature extraction, excel at capturing the intricate spatial information inherent in images, which is crucial for discerning complex cellular

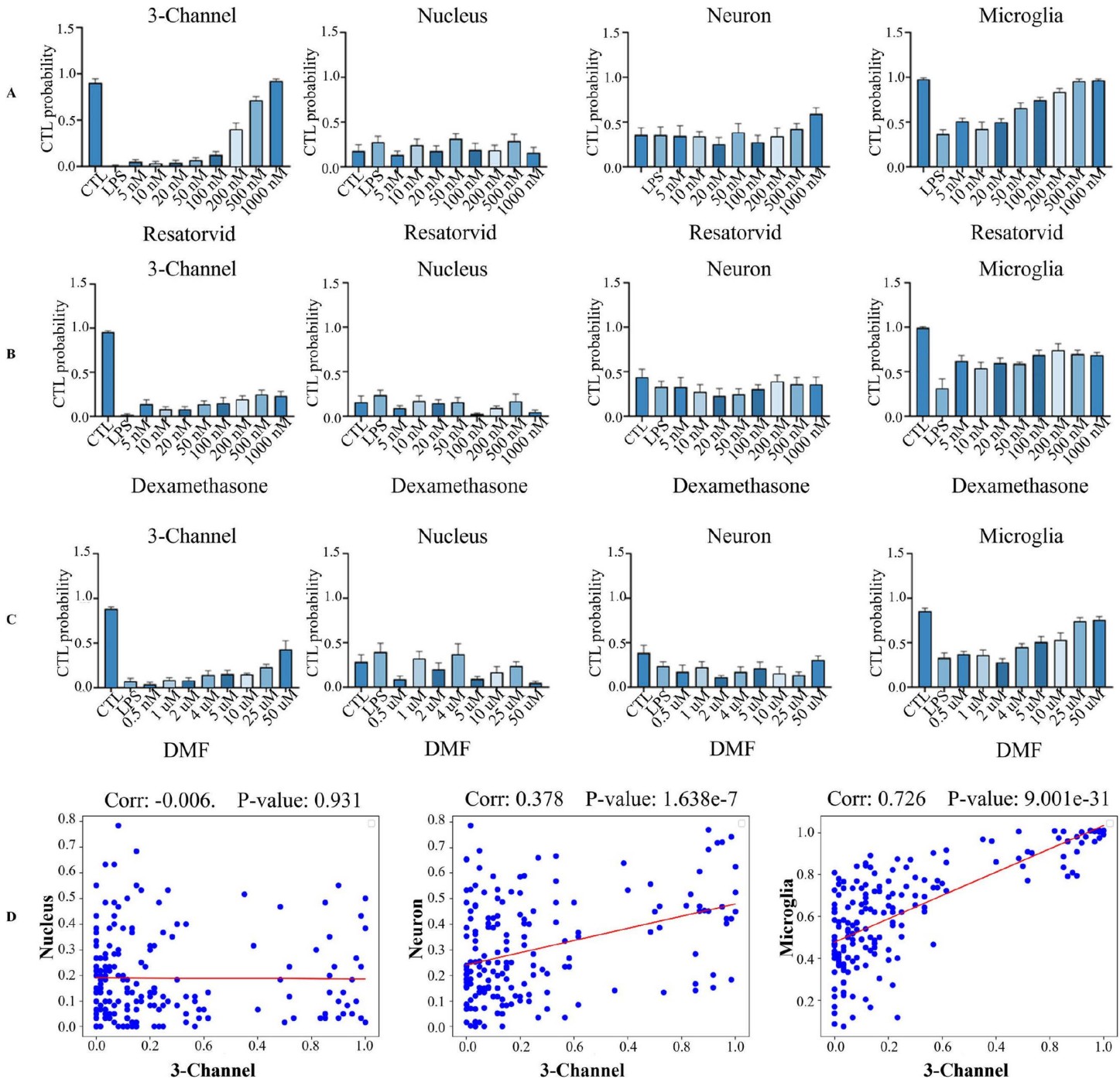

**Fig 6. Percentage of CTL prediction and analysis of correlation between predictions from three channels and those from each individual channel.** (A, B, C) Proportions of cell images classified as CTL following treatment with Resatorvid, Dexamethasone, and DMF. Error bars indicate the standard error of mean of these proportions across wells. (D) Similarity analysis with Pearson correlation to compare predictions from the combined three channels with those obtained from each individual channel. Each data point in the scatter plot represents the predicted control probability for each well.

phenotypes [32,33]. Conversely, ViT, leveraging long-range dependencies via self-attention mechanisms, showcases promising potential in comprehending entire image representations [15]. Our thorough investigation demonstrated that CNN-based models outperformed ViT in

tasks requiring detailed analysis of local features (Fig 3C). This was particularly evident when investigating microglia cells, which show diverse phenotypic states with distinct branching patterns and cell body shapes. This superiority is likely due to the spatial arrangements and local patterns that are crucial in defining cell images, with the architecture of CNN being well-suited to capturing the hierarchical nature of cellular features. Although the model showed a slightly lower F1 score at intermediate levels of the severity, it still consistently predicted the LPS concentrations that are closest to the actual dose administered to the cells (Fig 3D). Such performance emphasizes the model's ability to reliably approximate the cellular states across a spectrum of concentrations. In addition, the model exhibited remarkable performance in terms of robust predictions, achieving an F1 score of approximately 88%. This balanced measure of precision and recall demonstrates its robustness in the multi-classification task, which is specifically important in compound discovery where false predictions could lead to significant costs. Furthermore, when predicting in a well-wise manner (Fig 3E), the model showed superior performance in classifying all the pathological states.

Despite meticulous design, the application of CNNs to plate-cultured cell images revealed confounding plate effects akin to batch effects, exerting an unwanted influence on pertinent biological variables (S2 Fig) [34,35]. The batch effects can introduce an optimistic bias in validations and significantly decrease the model performance when applied to unseen data. This necessitated addressing these effects during the analysis. To employ an ensemble-based strategy (Fig 4B) for mitigating plate-level effects, we organized the training dataset into plate sets for the modified cross-validation (Fig 4A). Multiple models, fitted to different plate combinations, each generated a prediction on a cell image. We finally tested whether the ensemble model could accurately capture the dose-dependent effects of well-known anti-inflammatory compounds by estimating the unbiased measure of normal cell status. During the inference, the combined predicted values effectively smoothed biases caused by inherent batch effects (G in S2 Fig), resulting in superior performance compared to conventional approaches. In dealing with biological data, characterized by high variability and complexity, a single model may not be adequate in capturing all relevant information. This ensemble-based approach is an effective strategy in our study, enabling us to overcome the challenges posed by plate effects and reliably predict cellular states with improved F1 score and stability.

A detailed analysis of feature maps, intermediate outputs of CNNs, was conducted to enhance our understanding of how our model predicts pathological states. Analyzing feature maps extracted from various EfficientNet-B5 blocks revealed distinct patterns and essential characteristics for predictions [36]. These maps, derived from convolutional and activation layers, provide valuable insights into complex image data processing. The activation layer maps clearly showed desired cellular phenotypes, indicating efficient learning and capture of essential three-channel image characteristics (S1 Appendix and S3 Fig). Particularly, microglia feature maps displayed distinct distributions of feature importance (S1 Appendix and S4 Fig) in a way that emphasizes specific regions over others. This clarity enables the model to efficiently differentiate between relevant information and less relevant information for its classification. The near-uniform distributions of feature importance in the other two maps (neurons and nucleic acids) without emphasis on certain areas complicated the task of identifying which aspects of the cell images were critical to the prediction. These results strongly suggest the significant impact of the microglia on predictive performance.

Combining the images of three cellular markers into a unified image presents a compelling strategy, harnessing the strengths of individual maps while mitigating limitations. This combination provides the model with complementary information, enabling more informed decisions in tasks such as classification and recognition. Even with the possibility of feature redundancies, the simultaneous utilization of the three cellular markers proved advantageous,

improving the model's comprehension of cellular states, and potentially enhancing its general-izability on diverse datasets (Fig 6).

Our classification strategy employs distinct approaches for the model training and final compound screening. During training, the model was tasked with classifying six cellular states, enabling it to learn a comprehensive spectrum of inflammatory states and their corresponding morphological features. However, for the final compound screening phase, we adjusted the model to generate binary predictions (control vs. LPS-treated) and quantified the result as the percentages of cells predicted to be in a control state. This binary evaluation corresponds with the main goal of drug screening: determining whether a compound can restore the inflamed cells to control-like cells. Expressing the predictions as a percentage provides a quantitative and effective metric for assessing drug efficacy, facilitating the establishment of dose response relationships. This strategic transition from multi-class training to quantitative evaluation via binary predictions leverages the model's ability to discern subtle morphological changes.

While our approach demonstrates significant potential, there are opportunities for further refinement. The model's ability to capture intermediate levels of inflammation needs enhance-ments to capture subtle transitions between cellular states. While microglia markers demon-strated the strongest predictive signals in our analysis, the multi-channel integration helps validate that observed morphological changes are specifically related to inflammation (Fig 6). This approach could be optimized for different applications, potentially with varying weights assigned to each channel depending on the specific cellular responses. Increasing the dataset size would enhance generalizability across various experimental conditions, and modifying the model for other cell types might extend its applicability beyond primary cultured cells. Though our ensemble strategy effectively addresses plate-level effects, the approach introduced by this approach presents both advantages and challenges. While enhancing predictive power, the approach demands optimizing computational efficiency to further support high-throughput screening. ViT has demonstrated superior capability in detecting global dependencies within images, suggesting that the model trained on larger datasets enables better understanding of complex intercellular relationships. Therefore, although ViT models proved less optimal for our requirements, their capabilities, combined with CNN models' strength in local feature detection, point to promising direction for future work. Future studies should explore hybrid architectures that integrate the complementary proficiency of CNN and ViT models to enhance sensitivity to both morphological changes and intercellular relationships.

Our AI-derived screening is specifically designed to utilize morphological characteristics from microscopic images of neuronal and glial cells with a combination of fluorescent mark-ers. In developing the phenotypic cell-based model, we have employed primary cultured cells rather than cell lines. The use of cultured cells and the general molecular markers (neurons and microglia) for CNS provides unbiased cell profiles, more accurately reflecting neuroin-flammation compared to target-centered approaches [37]. The cutting-edge CNN-based algorithm leveraged the unbiased morphological information and intricate interactions in the cultured cells to efficiently and quantitatively evaluate both cellular states and the efficacy of known bioactive compounds. This innovative integration of the DL algorithm in cell-based assays can facilitate the rapid screening of thousands of candidate compounds during drug discovery and provide valuable insights into their mechanisms of action.

## Materials and methods

### Animal experiments

Mice for primary culture were purchased from KOATECH (Republic of Korea). All animal experiments were conducted according to the Ministry of Food and Drug Safety (MFDS)

guidelines for the maintenance and use of animals and were approved by the Institutional Animal Care and Use Committee (IACUC) of center at Woojung Bio (Republic of Korea, IACUC number: IACUC2004-036). Consistently, we performed all animal experiments in accordance with the ARRIVE (Animal Research: Reporting of In Vivo Experiments) guidelines.

## Primary cortical neuronal culture

Primary neuro-glia culture was prepared from E16 (embryonic day 16) C57/BL6 mice (KOAT-ECH). Pregnant mice were euthanized by carbon dioxide ($CO_2$) inhalation using compressed $CO_2$ gas and embryos were collected. Cortices from 5 embryos were dissected out and collected into a 15-ml conical tube containing 5-ml HEPES-buffered HBSS. After collecting the tissue sample at the bottom by gravity, supernatant was discarded, and 3-ml of 0.25% trypsin (Hyclone™, SH-30042.01) was added. Following a 15 min incubation with trypsin, supernatant was discarded, and 5-ml of DNase containing HBSS was added to dissociate cells. Cells were collected by centrifugation at 1,000 rpm for 3 min and resuspended with a 1:1 mixture of Neurobasal medium (Gibco, 21103-049) supplemented with 2% B-27 (Gibco, 17504-044), 2 mM L-glutamine (Sigma, G7513) and Minimum Essential Medium (MEM, Gibco, 11090-081) supplemented with 10% Fetal Bovine Serum (YOUNG IN, US-FBS-500), 10% Horse Serum (Gibco, 16050-122), 1× MEM-NEAA (non-essential amino acids, Gibco, 11140-050), 2 mM L-Glutamine, 1 mM Sodium pyruvate (Sigma, S8636), 0.2 g/ml D-glucose (Gibco, A2494001), 1% Penicillin-Streptomycin (Gibco, 15070063). The density of viable cells was counted using an automated cell counter with trypan blue stain, and cells were plated at $8 \times 10^4$ cells per well in Poly-D-Lysine (Sigma, P6407) and Laminin (Corning, 345232)-coated 96-well culture plates (Thermo, 167008). The media was changed every 2 to 3 days during the experiment.

Lipopolysaccharide (LPS, Sigma, L4391) stimulation and compound administration were conducted at five days after seeding. To acquire image sets for model learning, LPS was diluted into media for the final concentration of 0.005–20 μg/ml. To acquire image sets for model test, compounds were diluted into 0.005 μg/ml of LPS-containing media. Final concentration of compounds was 5–1,000 nM for resatorvid (ChemScene, CS-0408) and dexamethasone (ChemScene, CS-1505), or 0.5–50 μM for DMF (ChemScene, CS-0909). Two days after administration, cells were fixed for further immunostaining.

## Immunocytochemistry

Cells were fixed with 10% formalin solution for 20 min and permeabilized with 0.1% TritonX-100 in PBS. Cells were blocked with LICOR Odyssey blocking buffer (LI-COR, 927-60003) for 1 hr at room temperature. Primary antibodies were diluted into blocking buffer as follows: 1:2,000 for mouse anti-MAP2 (microtubule associated protein 2) antibody (Milipore, MAB3418), 1:1,000 for rabbit anti-Iba-1 (ionized calcium-binding adapter molecule 1) antibody (Abcam, ab178846). Primary antibodies were incubated overnight at 4°C. After washing with 0.1% Tween-20 in PBS, samples were incubated with secondary antibody solution for 2 hr at room temperature. Secondary antibodies were diluted into 0.1% Tween-20 in PBS as follows; 1:1,000 for AlexaFluor-488 conjugated goat anti-mouse IgG antibody (Invitrogen, A-10680) and AlexaFluor-594 conjugated goat anti-rabbit IgG antibody (Invitrogen, A32740). Nuclei were counterstained with DAPI (4′,6-diamidino-2-phenylindole, Thermo, D1306) for 15 min at room temperature.

## Image acquisition

Fluorescent images were acquired using a Cytation 5 Automated Imaging Multi-Mode reader (BioTek). A total of nine montage images (3×3) per well were acquired using 10× objective

lens and the area of each image was $1,389 \times 1,389$ in μm ($1,992 \times 1,992$ in pixel). Nuclei, MAP2, and Iba-1 images were acquired using DAPI, GFP, and Texas Red® filter channels, respectively.

We used a total of 3,150 cellular images collected from seven experimental plates to train CNN and ViT models. Each plate contained six experimental classes: one control group and five groups treated with different LPS concentrations. For each class, we used five replicate wells, with each well generating 15 cropped images. Thus, each plate yielded 450 images (6 classes × 5 wells × 15 crops = 450 images), resulting in a total dataset of 3,150 images across all seven plates (450 images × 7 plates).

## Image stitching and cropping

Nine images from a well obtained from the image acquisition step were combined to create a single stitched image. To reduce noise and enhance generalization performance, stitched images were cropped into a $4 \times 4$ grid format, typically $2,400 \times 2,400$ pixels, with an overlap between adjacent crops. This process produces 16 cropped images per well. The cropped image located in the bottom-right corner was discarded due to noise issues. As illustrated in Fig 2B, 15 cropped images are utilized for each well.

## Image annotation

Cropped images in the training dataset were annotated with the corresponding LPS concentration treated to each well.

## Data normalization and augmentation

Before the model training, the obtained cropped data underwent normalization and various augmentation techniques. Normalization involved calculating the mean and standard deviation for each channel across all the images in the training dataset. This normalization step established the foundation for consistent and standardized input across the dataset. For each channel, the normalization step can be expressed formally as equation (2).

$$\tilde{p}_{ij} = \frac{p_{ij} - \mu \times 255}{\sigma \times 255} \tag{2}$$

Where $p_{i,j}$ is an original pixel value from $i^{th}$ row and $j^{th}$ column of an image ranging from 0 to 255, and $\tilde{p}_{i,j}$ is a normalized pixel value of $p_{i,j}$. The symbols $\mu$ and $\sigma$ denote the mean and standard deviation, respectively, derived from the training dataset. They are computed after adjusting the original pixel values of the images, which range from 0 to 255, to a scale where each pixel value falls within the range of 0 to 1.

In addition to normalization, various augmentation techniques were applied to augment the dataset and improve the model's robustness. A diverse set of augmentation methods was randomly employed, encompassing horizontal and vertical flips, random rotation by 90 degrees, solarization, multiplicative noise, RGB shift, motion blur, optical distortion, Gaussian noise, Gaussian blur, random sun flare, random snow, sharpening, embossing, and further normalization. This comprehensive set of augmentations aimed to expose the model to a wide range of variations, enhancing its ability to generalize and make accurate predictions on diverse inputs.

## Preprocessing

In this paper, we recognized the significance of preprocessing in enhancing the performance of our deep learning model, particularly given the diverse characteristics of image data acquired under various conditions. The inherent variability in image brightness, color, size,

and the presence of noise or distortion pose challenges to effective model learning. Therefore, preprocessing steps were carefully designed to transform the data into a format conducive to robust model training.

Preprocessing was conducted as a pivotal preparatory step before engaging in model training. The tasks were organized into three main categories: data augmentation, data conversion, and data normalization.

To mitigate overfitting and enhance data diversity, various augmentation techniques, such as shift, rotation, flip, enlargement, and reduction of image coordinates, were applied to approximately 10% of the dataset [24,25]. These transformations aimed to simulate different conditions and perspectives, thereby enriching the dataset.

In data conversion, various methods were used to improve model performance by changing data characteristics. In the case of solarization, the effect of increasing contrast is by taking advantage of the phenomenon in which the bright part of the image turns black when the brightness of the image is adjusted above a certain level, as described in Equation (3)

$$S(I) = I^* (I > 0.95) \tag{3}$$

Multiplicative noise, as shown in Equation (4) is a preprocessing method that adds noise by multiplying each pixel of the image by a random value. It can increase the diversity of the image and improve the generalization performance of the model.

$$N(I) = I^* (1 + noise) \tag{4}$$

RGB shift is a preprocessing method that shifts the value of each channel of the image by a certain range, and can improve model performance by adjusting the brightness, contrast, and color of the image, among other techniques [38,39].

To ensure consistent and efficient model training, data normalization was performed by standardizing the pixel values. This step was crucial for preventing potential issues related to varying scales and distributions in the input data.

Our choices in preprocessing techniques were guided by their effectiveness in addressing overfitting, enhancing data diversity, and improving model generalization. Furthermore, the specific parameters for each technique were selected based on careful consideration of their impact on the data and model performance. The sequence of preprocessing steps was also thoughtfully arranged to maximize their collective benefit.

## Model evaluation

To identify the most effective model for classifying cell inflammation images, we conducted an extensive evaluation using a diverse set of CNN and ViT architectures. The models considered include Inception_v3 [40], ResNet50 [41], DenseNet201 [42], Reg-Net_Y_3_2GF [32], and EfficientNet series (B0 to B5) [43], as well as ViT-based models such as ConvNeXt_Tiny [44], MaxViT_T [45], and Swin_V2_T [46]. The number of parameters varied across these architectures, with Inception_v3 featuring 27.2M parameters, ResNet50 with 25.6M, DenseNet201 with 20.0M, RegNetY-3.2GF with 19.4M, and EfficientNet-B0 with 5.3M.

While CNN learns from 15 cropped images per well, ViT, as depicted in Fig 3B, divides a single cropped image into nine parts and flattens them for sequential training. Consequently, ViT learns from a total of 135 images per well.

In assessing model performance, accuracy and macro F1 score, which can be expressed as equations (5) and (6), are the primary metrics. While other metrics were considered, accuracy and macro F1 score were deemed most relevant for the task of cell inflammation image classification. This choice aligns with the ultimate goal of achieving a model that can effectively distinguish between different inflammatory states in cellular images.

$$Accuracy = \frac{Number\ of\ Correct\ Predictions}{Number\ of\ Total\ Predictions} \tag{5}$$

$$F1_{macro} = \frac{1}{N}\sum_{i=1}^{N}\frac{2 \times TP_i}{2 \times TP_i + FN_i + FP_i} \tag{6}$$

Where $N$ is the number of categories, $TP_i$, $FN_i$ and $FP_i$ are respectively true positive, false negative, and false positive for category $i$.

## Model training

To leverage the powerful features of EfficientNet-B5, a transfer learning approach was employed. The model was initialized with pre-trained weights derived from a comprehensive dataset, namely ImageNet. These pre-trained weights provide the model with a robust initial set of feature extractors, enhancing its capacity to discern both low-level and high-level patterns within images.

Fine-tuning is a crucial step in adapting EfficientNet-B5 to the specific task of classifying inflammatory cell images. This process involves adjusting the model's weights based on our dataset while preserving, to some extent, the knowledge gained from pre-training on ImageNet. The fine-tuning procedure aimed to enhance the model's ability to discriminate among inflammatory cell images induced by LPS. This tailored approach ensured that the model was finely tuned to the specific features and subtle differences of the dataset, optimizing its performance for the targeted classification task.

During the training phase, our model was configured with specific settings to optimize its performance. A batch size of 8 was chosen to strike a balance between efficient memory usage and computational efficiency. The optimizer employed was stochastic gradient descent, complemented by a learning rate scheduler that gradually decreased the learning rate over time to ensure stable convergence. Every five epochs, this scheduler decreases the learning rate by a factor of 0.25. To tailor the model for multi-class classification, a categorical cross-entropy loss function was used.

To enhance computational efficiency and memory usage, we employed the mixed precision [47] technique during model training. This method leverages both 16-bit and 32-bit floating-point representations to optimize computational speed. The adoption of mixed precision played a crucial role in achieving a streamlined training process, making it more resource efficient.

## Session information

This entire process of this research, from data preprocessing to image classification, was conducted using Python (version 3.10.6) open-source tools. The Python packages used in this study include Torch [20], NumPy [48], pandas [49], Matplotlib [50], scikit-learn [51], OpenCV-python [52], Ray [53], and albumentations [54]. The specific versions of each package used in the research were as follows: torch 2.0.0, NumPy 1.21.5, pandas 2.0.1, Matplotlib 3.7.1, scikit-learn 1.2.2, OpenCV-Python 4.7.0.72, ray 2.4.0, and albumentations 1.3.0. These packages can be installed using the `pip3 install package name==version`.

## Supporting information

**S1 Appendix. Additional information about ensemble approach and feature map.**
(DOCX)

**S1 Fig. Representative microscopic images of cells stained with Iba-1 (microglia, red).**
MAP-2 (neuron, green), and DAPI (nucleic acids, blue). The first and second rows show the cells treated with increasing concentrations of LPS. The third row displays the cells exposed to LPS and a range of anti-inflammatory compounds, providing a comparative perspective on the effectiveness of these compounds.
(TIF)

**S2 Fig. Ensemble-based approach to overcome plate effects.** (A) The confusion matrix from the model deployment step illustrates how accurately four models classify the LPS concentration given to the primary cultured cells. The diagonal entries of the matrix represent the number of the correctly classified wells of plates. (**B-F**) In the efficacy test, ratios of cell images classified as control are expressed as a function of increasing concentrations of an anti-inflammatory compound. The ratios for 4 models (**B-E**) derived from 4-fold cross-validation alongside the ensemble model (**F**). (**G**) A heat map of Pearson correlations between the control probabilities and the doses of the three anti-inflammatory drugs administered to the primary cultured cells.
(TIF)

**S3 Fig. Feature Map Analysis of Different Cellular Phenotypes Extracted from the Convolutional Layer.** This figure presents feature maps extracted from the Convolutional Layer for various cellular properties. The Convolutional Layer captures low-level features in images, and this figure visualizes the structural characteristics identified for each cellular marker. In the context of the viridis colormap, which is often used for this purpose, the spectrum typically ranges from purple to yellow. Purple or dark colors represent areas with lower weights or less significance in the feature map, indicating regions that the model does not consider important for its predictions. On the other hand, yellow or bright colors correspond to higher weights or more important regions, showing features that the model heavily relies on for its decision-making process. (**A**) Nucleic acids (blue): The map emphasizes subtle features such as nuclear structures and arrangements within cells, clearly differentiating areas with and without nucleic acids. (**B**) Neurons (green): The complex morphology and network of neurons are reflected, with distinct visualization of cell bodies and dendritic structures. (**C**) Microglia (red): The unique shape and distribution of microglia are highlighted, visualizing structural changes based on the cell's activity state.
(TIF)

**S4 Fig. Feature Map Analysis of Different Cellular Phenotypes Extracted from the Activation Layer.** This figure demonstrates feature maps from the Activation Layer, illustrating which regions the model deems significant. Using the viridis colormap, areas of importance are differentiated by color, with each feature map showing the following characteristics. (**A**) Nucleic acids (blue): There is a clear distinction between important and less significant regions, with vital parts of nucleic acids highlighted in brighter colors. (**B**) Neurons (green): The map emphasizes significant structures of neurons, though the differentiation of importance may be less distinct. (**C**) Microglia (red): Key areas of activated microglia are brightly colored, effectively revealing important features.
(TIF)

## Author contributions

**Conceptualization:** Hyunseok Bahng, Jung-Pyo Oh, Sungjin Lee, Jaehong Yu, Jongju Bae, Eun Jung Kim, Sang-Hun Bae, Ji-Hyun Lee.

**Data curation:** Jung-Pyo Oh, Eun Jung Kim.

**Formal analysis:** Hyunseok Bahng, Sungjin Lee, Jaehong Yu, Jongju Bae.

**Investigation:** Hyunseok Bahng, Sungjin Lee, Jaehong Yu, Eun Jung Kim, Sang-Hun Bae.

**Methodology:** Jung-Pyo Oh, Eun Jung Kim.

**Software:** Hyunseok Bahng, Sungjin Lee, Jaehong Yu, Jongju Bae.

**Supervision:** Ji-Hyun Lee.

**Visualization:** Hyunseok Bahng.

**Writing – original draft:** Hyunseok Bahng, Jung-Pyo Oh, Sungjin Lee, Jaehong Yu.

**Writing – review & editing:** Sang-Hun Bae, Ji-Hyun Lee.

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
