## [Decision Letter · Decision Letter 0]

8 Nov 2024

PONE-D-24-44853Unveiling CNS Cell Morphology with Deep Learning: A Gateway to Anti-Inflammatory Compound ScreeningPLOS ONE

Dear Dr. Bae,

Thank you for submitting your manuscript to PLOS ONE. After careful consideration, we feel that it has merit but does not fully meet PLOS ONE’s publication criteria as it currently stands. Therefore, we invite you to submit a revised version of the manuscript that addresses the points raised during the review process.

If the reviewers have suggested specific citations to be added during revision, please feel free to decide if the same is needed and adds value to the article or no. 

We look forward to receiving your revised manuscript.

Kind regards,

Ghulam Md Ashraf, Ph.D.

Academic Editor

PLOS ONE

2. Thank you for uploading your study's underlying data set. Unfortunately, the repository you have noted in your Data Availability statement does not qualify as an acceptable data repository according to PLOS's standards. At this time, please upload the minimal data set necessary to replicate your study's findings to a stable, public repository (such as figshare or Dryad) and provide us with the relevant URLs, DOIs, or accession numbers that may be used to access these data. For a list of recommended repositories and additional information on PLOS standards for data deposition, please see https://journals.plos.org/plosone/s/recommended-repositories.

Additional Editor Comments (if provided):

Reviewers' comments:

Reviewer's Responses to Questions

**Comments to the Author**

1. Is the manuscript technically sound, and do the data support the conclusions?

Reviewer #1: Yes

Reviewer #2: Partly

Reviewer #3: Partly

Reviewer #4: No

Reviewer #5: Yes

2. Has the statistical analysis been performed appropriately and rigorously? 

Reviewer #1: Yes

Reviewer #2: No

Reviewer #3: No

Reviewer #4: No

Reviewer #5: N/A

3. Have the authors made all data underlying the findings in their manuscript fully available?

Reviewer #1: Yes

Reviewer #2: Yes

Reviewer #3: Yes

Reviewer #4: Yes

Reviewer #5: Yes

4. Is the manuscript presented in an intelligible fashion and written in standard English?

Reviewer #1: Yes

Reviewer #2: Yes

Reviewer #3: Yes

Reviewer #4: No

Reviewer #5: Yes

5. Review Comments to the Author

Reviewer #1: Bahng et al. present a robust DL-based image analysis pipeline for classifying CNS cell morphology in the context of inflammation. The authors present an effective study design of dosing LPS to correspond with increasing inflammatory response in the cells, eliminating the need for tedious and often subjective expert-labelling of ground truth required in model development. The authors also present a novel leave-one-plate-out validation procedure used in their model development process that improves generalizability of their models, alongside extensive data transformations and preprocess to increase robustness. Some methodological clarifications, outlined below, would benefit the manuscript:

While the model development process is very well-articulated, the model evaluation portion is less clear. Namely, the paragraph beginning on line 209 references two confusion matrices (Figure 3c and 3d) implying a testing paradigm on 5 wells per class (Fig 3d) and their constituent 15 cropped images for a total of 75 image-wise predictions per class (Fig 3c). It is unclear where these 5 wells were selected from (i.e., from the held-out validation fold?). Lines 251-261 under section “Ensemble-based batch effect mitigation” mentions “predicting one of the 15 cropped images generated from a single well” which does not help in clarifying the confusion regarding the model testing procedure.

Figure 3 on Line 175 seems to have replicated subfigures c-e.

Regarding the leave-one-plate-out cross-validation, the authors chose not to select an optimal validation model and instead combine all trained models with an ensemble based-strategy. They claim that this reduces batch-like effects from the plates. However, it is unclear why the authors chose to still group plates together into 4 folds (combining two plates per fold except the final fold which includes only one plate) and not instead run 7 trials with 7 folds corresponding to each distinct plate.

The dose-dependent effects of anti-inflammatory compounds, particularly in microglial cells is impressive. However, the inability of the model to predict controls as such in the nucleus and neuron channels is concerning (<50%). Further explanation from the authors regarding the poor performance in these cell types, especially with the non-inflamed states, would benefit the discussion section of the manuscript. The authors claim themselves that “The near-uniform distributions of feature importance in the other two maps (neurons and nucleic acids) without emphasis on certain areas complicated the task of identifying which aspects of the cell images were critical to the prediction” (lines 358-360), yet continue to support the stance of combining the three images of cellular markers into a unified image, despite most of the contributions seemingly coming from the microglial cells (i.e., microglia-channel only models seem to show anti-inflammatory efficacy more clearly than the 3-channel models in Figure 5). A more thorough discussion on what the other two cell types contribute may help strengthen this stance.

Regarding data preprocessing, the authors highlight the impacts of preprocessing parameters on model performance (lines 492-496) but do not explain further how these parameters were selected. A section in the supplementary data addressing this parameter selection would be beneficial.

Overall, the authors’ undertaking of an intensive comparative study on 14 different models with two very relevant architectures (CNNs and ViTs) is commendable. The conclusions support the well elucidated rationale for phenotypic screening of anti-inflammatory compounds in the context of pharmacological testing. Only a few methodological clarifications are needed before publication.

Reviewer #2: -Figure 1: You label part A with "Test image dataset" but call it the "Validation set" in part B, as well as in the rest of the document. I would change the former to say Validation to be more consistent, because you do not have both a Validation set and a Test set.

-Line 114: Can you specify the exact size of the training set? It is too vague to say "thousands" and would be helpful for context as to how many are in each class and why you would need augmentation.

-Line 167: Why was only 10% of the dataset augmented?

-Line 172: Can you quantify what the positive effect of preprocessing on the accuracy was?

-Figure 3: In my version, 3C-E are duplicated. Please remove one set.

-Line 203: Is the accuracy based on the image task or the well task? It is not explained here. Are different models better for different tasks?

-Line 206: Typo "CNNs models" -> "CNN models"

-Figure 4: Typo "Model devlopment set" -> "Model development set"

-Line 240: It is more rigorous to say that it is a form of stratified cross-validation because you have intentionally split your data to be ordered in some way. I would mention that is also a standard method of splitting for cross-validation in addition to random.

-Line 258: There is confusion because you suddenly introduce this binary variable R when you previously had a classification with multiple classes. It is not explained until the next section why this is the case, so you may want to include some of that background after you introduce R.

-Figure 5: The error bars are not well-explained. Can you provide some more details there?

-Line 275: If the ultimate goal was to train a model for binary classification, why first train models to classify various degrees? It is not well explained here.

-The Discussion section reads more like a Conclusion section. It would be helpful to distinguish the Discussion from the Conclusion here.

-Line 321: Typo "CNNs-based models" -> "CNN-based models"

-Line 328: Earlier, you noted 88% as the accuracy, but now it is noted as the F1 score. Later in the Methods section, you distinguish between accuracy and F1 score, so which one is actually 88%? And why was the F1 score not mentioned earlier?

-Line 329: You mention the importance in the multi-class task, but the Results focused on the binary classification. The message is a little bit unclear to me.

-Line 333: Reword "CNNs application" -> "application of CNNs"

-Line 338: Again, it is not modified cross-validation, just stratified.

-Line 339: Based on your approach, it is more rigorous to say multiple models each generated a prediction, rather than each model generating multiple predictions.

-Line 344: Where is this the prevailing notion? I feel most people in the machine learning field would not jump to such a conclusion.

-Line 355: Why are those figures in Supplementary rather than in the main text? They seem important to note.

-Line 373: Typo "CNNs-based algorithm" -> "CNN-based algorithm"

-Line 499: It does not contribute because it mentions but does not quantify the benefits on accuracy.

-Did you consider the classification confidence level? For example, in a binary case, does your model return a probability of the classes, and are some of these inferences more confident than others?

Reviewer #3: This manuscript presents a promising DL-based framework for CNS cell morphology analysis in drug screening. The manuscript is well written. However, with clarifications on the methodology, additional visualizations, and expanded discussions on applications and limitations, this work could make a strong contribution to the field. Here are some comments and suggestions to enhance the quality and impact of the work:

1. The manuscript discusses the use of CNN and ViT for classifying inflammatory states in CNS cells. While EfficientNet-B5 is selected as the optimal model, the criteria and reasoning behind the preference over ViT models could be further clarified. Including more detailed comparisons, particularly around model interpretability and applicability for CNS cell morphology, could enhance the reader’s understanding.

2. How many images are in the training and test set? I suggest that the authors clearly state the number of training and test samples and the train-test-split ratio.

3. The confusion matrix in Figure 3 is not very clear. Which model is used to generate the plot and why are there few sample sizes?

4. The ensemble method adopted to mitigate batch effects is a strong feature of this study. However, additional clarification on the effectiveness of the leave-one-plate-out cross-validation approach would be beneficial. Specifically, statistical comparisons between model performance with and without this approach could provide quantitative support for its utility.

5. Including visualizations, such as t-SNE plots, to demonstrate how the ensemble approach reduces batch effect in feature space would make this approach clearer and support its validity.

6. There are a few grammatical errors on line 228 and 229. I suggest that the authors thoroughly read through the manuscript and make correct all errors.

7. I suggest that the figures in the paper should be enhanced for clarity.

8. I suggest that the authors number all the equations in the paper.

8. I suggest that the authors add a brief section discussing potential limitations, particularly related to dataset size, generalizability, and model complexity.

Reviewer #4: The study presented lacks sufficient depth in addressing the existing challenges and limitations of deep learning (DL)-based image analysis in neuropathological contexts. While the authors mention the issues of labeled data requirements, detection of subtle cellular changes, and batch effects, the manuscript does not provide substantial evidence that these issues have been adequately overcome. Additionally, the study's approach appears to rely on in-house data, which could limit its generalizability and broader applicability in neuroinflammation research. The manuscript’s description of “enhancing understanding” and “streamlining processes” remains vague, with no clear indication of how these improvements quantitatively advance current methodologies. Furthermore, there is limited information on how the DL model was optimized for detecting morphological phenotypes specific to neuronal and glial cells, especially given the inherent complexity and variability within CNS cell types. Finally, the study lacks rigorous validation steps or comparative analysis with existing methods, raising questions about the reproducibility and robustness of the findings.

Reviewer #5: The authors present a compelling study that utilizes deep learning (DL) to analyze CNS cell morphology, focusing on screening anti-inflammatory compounds.This study introduces a novel method for cell morphology analysis, particularly in handling batch effects through ensemble modeling and applying DL to phenotype-based drug screening.

1) The methods was validated with cross validation but the authors didn't discuss how would such system be deployed for real-wrold screening especially on the compute resource requirement.

2) The exploration of data augmentation to address the batch effect is lacking, the use of ensemble method is useful but other approaches can be explored as part of the comparison.

3) Model interoperability study is also missing, the authors can explore how to interpret the proposed CNN based model and highlight the feature importance through attribution via methods such as saliency and guided gradient.

4) The integration of other real-time phenotypes in addition to imaging input can potentially improve model capacity.

6. PLOS authors have the option to publish the peer review history of their article (what does this mean? ). If published, this will include your full peer review and any attached files.

**Do you want your identity to be public for this peer review?** For information about this choice, including consent withdrawal, please see our Privacy Policy .

Reviewer #1: **Yes: ** Earvin S. Tio

Reviewer #2: No

Reviewer #3: No

Reviewer #4: **Yes: ** Sachchida Nand Rai

Reviewer #5: No

---

## [Author Response · Author response to Decision Letter 1]

23 Dec 2024

We thank you and the reviewers for your careful evaluation of our manuscript “Unveiling CNS cell morphology with deep learning: A gateway to anti-inflammatory compound screening”. We appreciate the constructive feedback, which has helped us improve our manuscript significantly.

We have thoroughly addressed all the comments and revised our manuscript accordingly. Below, we provide a point-by-point response to each reviewer's comments.

Reviewer #1:

Comment 1:

While the model development process is very well-articulated, the model evaluation portion is less clear. Namely, the paragraph beginning on line 209 references two confusion matrices (Figure 3c and 3d) implying a testing paradigm on 5 wells per class (Fig 3d) and their constituent 15 cropped images for a total of 75 image-wise predictions per class (Fig 3c). It is unclear where these 5 wells were selected from (i.e., from the held-out validation fold?). Lines 251-261 under section “Ensemble-based batch effect mitigation” mentions “predicting one of the 15 cropped images generated from a single well” which does not help in clarifying the confusion regarding the model testing procedure.

Response to Comment 1:

Thank you for pointing out the need for clarity in our model evaluation procedure.

The confusion matrices in Figure 3c and 3d were generated using the best performing model among the four EfficientNet-B5 models trained during our leave-plates-out cross-validation process. Specifically, these results came from the validation case where the validation set contained one plate.

For this single validation plate:

1. Image-level predictions (Figure 3d):

o Contains 450 images total (5 wells × 15 cropped images × 6 classes)

o Shows how individual cropped cell images were classified

2. Well-level predictions (Figure 3e):

o Contains 30 wells total (6 classes × 5 replicate wells)

o Represents how these wells were classified based on their inflammatory states

o Each well's classification is determined by aggregating predictions from its 15 cropped images

We have revised the Results section (line 245) to clearly explain this evaluation process and its relationship to our cross-validation procedure.

Comment 2:

Figure 3 on Line 175 seems to have replicated subfigures c-e.

Response to Comment 2:

Thank you for bringing this to our attention. Upon checking, we found that subfigures c-e were indeed mistakenly replicated in Figure 3. We have corrected Figure 3 to show each subfigure only once.

Comment 3:

Regarding the leave-one-plate-out cross-validation, the authors chose not to select an optimal validation model and instead combine all trained models with an ensemble based-strategy. They claim that this reduces batch-like effects from the plates. However, it is unclear why the authors chose to still group plates together into 4 folds (combining two plates per fold except the final fold which includes only one plate) and not instead run 7 trials with 7 folds corresponding to each distinct plate.

Response to Comment 3:

Thank you for this insightful question about our cross-validation design.

While using 7 folds (one for each plate) might appear more intuitive, we opted for 4 groups to optimize computational efficiency. This approach reduced computational overhead by training fewer models while still effectively capturing plate-to-plate variations. The reduced computation time is particularly crucial for practical drug screening applications, where rapid processing of large datasets is essential.

Comment 4:

The dose-dependent effects of anti-inflammatory compounds, particularly in microglial cells is impressive. However, the inability of the model to predict controls as such in the nucleus and neuron channels is concerning (<50%). Further explanation from the authors regarding the poor performance in these cell types, especially with the non-inflamed states, would benefit the discussion section of the manuscript. The authors claim themselves that “The near-uniform distributions of feature importance in the other two maps (neurons and nucleic acids) without emphasis on certain areas complicated the task of identifying which aspects of the cell images were critical to the prediction” (lines 358-360), yet continue to support the stance of combining the three images of cellular markers into a unified image, despite most of the contributions seemingly coming from the microglial cells (i.e., microglia-channel only models seem to show anti-inflammatory efficacy more clearly than the 3-channel models in Figure 5). A more thorough discussion on what the other two cell types contribute may help strengthen this stance.

Response to Comment 4:

Thank you for these insightful observations. We have added more explanation of the multi-channel integration to the Discussion section (line 497).

Comment 5:

Regarding data preprocessing, the authors highlight the impacts of preprocessing parameters on model performance (lines 492-496) but do not explain further how these parameters were selected. A section in the supplementary data addressing this parameter selection would be beneficial.

Response to Comment 5:

Thank you for raising this important point about our preprocessing parameter selection. We added explanations of each preprocessing method and its intended purpose. The unsupported claim about preprocessing impacts on model performance was removed (line 201). We followed standard preprocessing approaches rather than conducting systematic parameter optimization studies, so we chose not to add a supplementary section.

Comment 6:

Overall, the authors’ undertaking of an intensive comparative study on 14 different models with two very relevant architectures (CNNs and ViTs) is commendable. The conclusions support the well elucidated rationale for phenotypic screening of anti-inflammatory compounds in the context of pharmacological testing. Only a few methodological clarifications are needed before publication.

Response to Comment 6:

We have added more explanations to Discussion.

Reviewer #2:

Comment 7:

-Figure 1: You label part A with "Test image dataset" but call it the "Validation set" in part B, as well as in the rest of the document. I would change the former to say Validation to be more consistent, because you do not have both a Validation set and a Test set.

Response to Comment 7

Thank you for noting this terminology inconsistency. We have revised the label from "Test image dataset" to "Validation set" to maintain consistent terminology throughout the manuscript and figures.

Comment 8:

-Line 114: Can you specify the exact size of the training set? It is too vague to say "thousands" and would be helpful for context as to how many are in each class and why you would need augmentation.

Response to Comment 8:

We have integrated the following description of the number of samples (image-level, well-wise) to the method section (line 577) and clarified the reason behind using augmentation to “Transforming Cellular Microscopic Images for Deep Learning: Staining, Augmentation, and Normalization Techniques” section in Method.

Dataset Composition:

We used a total of 3,150 cellular images collected from seven experimental plates to train CNN and ViT models. Each plate contained six experimental classes: one control group and five groups treated with different LPS concentrations. For each class, we used five replicate wells, with each well generating 15 cropped images. Thus, each plate yielded 450 images (6 classes × 5 wells × 15 crops = 450 images), resulting in a total dataset of 3,150 images across all seven plates (450 images × 7 plates).

Cross-validation Design:

To ensure robust model validation while accounting for plate-specific effects, we implemented a modified leave-one-group-out cross-validation strategy. The seven plates were systematically combined into four groups. Three groups were formed by combining two plates, where one group was left with a single plate. During cross validation, one group served as the validation set while the remaining group formed the training set. This approach ensured that cell images from the same plate remined together in the same fold, preventing data leakage. The validation set contained either 900 images when using groups 1-3 (2 plates × 450 images per plate) or 450 images when using group 4 (1 plate × 450 images).

Well-wise prediction:

For well-wise predictions, we used the data at the well level across our experimental dataset. Each plate contained 30 wells, comprised of 6 classes (one control and five LPS concentrations) with 5 replicate wells per class. Across all seven plates, this yielded a total of 210 wells (30 wells × 7 plates).

Comment 9:

-Line 167: Why was only 10% of the dataset augmented?

Response to Comment 9:

Thank you for this important question about our data augmentation strategy. While augmentation is beneficial for mitigating overfitting, excessive augmentations may introduce noises that diverge from biological characteristics of the original data, potentially impacting model performance. We acknowledge that our choice of 10% was not determined empirically. However, by limiting augmentation to this level, we aimed to diversify the data while minimizing the risk of over-generalization. Given that this choice yielded good model performance, we did not need to empirically determine the optimal augmentation.

We have added the following explanations (line 188) to clarify their intended purpose

““Approximately 10% of the dataset was augmented using shift, rotation, flip, enlargement, and reduction of image coordinates, to increase dataset size and variability (Fig 2c, d). This level of augmentation was selected to balance enhancing the model’s robustness and preserving the integrity of the original data. While augmentation is beneficial for mitigating overfitting, excessive augmentations may introduce noises that diverge from biological characteristics of the original data, potentially impact model performance. By limiting augmentation to 10%, the data was diversified while minimizing the risk of over-generalization.”

Comment 10:

-Line 172: Can you quantify what the positive effect of preprocessing on the accuracy was?

Response to Comment 10:

Thank you for this important question. While we applied conventional preprocessing methods that are widely established in deep learning literature to improve model generalization, we acknowledge that we did not conduct studies to quantify the specific impact of each preprocessing step on our model's accuracy. This is a valuable suggestion for future work, where systematic evaluation of each preprocessing component's contribution could provide insights into their relative importance for cell morphology analysis.

We have removed the claim about the positive effects on accuracy (line 201)

Thank you for this suggestion.

Comment 11:

Figure 3: In my version, 3C-E are duplicated. Please remove one set.

Response to Comment 11:

We have fixed it. Thank you.

Comment 12:

-Line 203: Is the accuracy based on the image task or the well task? It is not explained here. Are different models better for different tasks?

Response to Comment 12:

The accuracy reported in our study is primarily based on the image-level prediction, which serves as an intermediate step toward well-level prediction. Specifically, the model predicts the class for each cropped image, and these predictions are then aggregated to determine the final prediction for the corresponding well.

The ultimate goal of our study is to accurately predict the state of each well, as this is critical for evaluating the drug’s efficacy. Well-level prediction provides a more robust and interpretable assessment of drug effects because it reflects the overall state of the cellular population within a given well, rather than individual images, which may exhibit variability.

Regarding the suitability of different models for different tasks, our findings indicate that the same model architecture can be effective for both image-level and well-level predictions when paired with an appropriate aggregation strategy. By using the ensemble method and the CTL proportion-based approach, we successfully bridged the gap between these two levels, achieving robust performance in well-level prediction.

The following are several benefits of well-wise predictions particularly for drug screening applications.

1. Enhanced Robustness: Well-wise predictions capture the collective state of cellular populations by aggregating multiple images per well, smoothing out variability in individual image predictions and better reflecting the biological reality of non-uniform cellular responses.

2. Improved performance: In Figure 3e, our model demonstrated superior classification accuracy across all pathological states when using well-wise predictions compared to image-wise predictions (Figure 3d).

3. Since compounds are administered at the well level, well-wise predictions correspond better with actual drug screening. This approach minimizes inaccurate predictions that could lead to significant costs in drug discovery processes.

4. Batch effect mitigation: The considerations of multiple images per well for a final prediction helps account for plate-to-plate variations.

5. Quantitative assessment: This quantitative assessment provided better insights into the degree of drug response by using the percentages of control cells in well.

Comment 13:

-Line 206: Typo "CNNs models" -> "CNN models"

Response to Comment 13:

We have fixed it.

Comment 14

-Figure 4: Typo "Model devlopment set" -> "Model development set"

Response to Comment 14:

Thank you for catching this typographical error. We have corrected "Model devlopment set" to "Model development set" in Figure 4 legend.

Comment 15

-Line 240: It is more rigorous to say that it is a form of stratified cross-validation because you have intentionally split your data to be ordered in some way. I would mention that is also a standard method of splitting for cross-validation in addition to random.

Response to Comment 15:

Thank you for this suggestion. While stratified cross-validation maintains similar class distribution across all folds, our approach follows a different principle - keeping all data from a single plate together in the same fold. This plate-based strategy was designed to reflect real-world conditions where new plates would be tested on models trained on different plates. While each plate in our experimental setup naturally contains a balanced class distribution, this balance comes from the experimental design itself, not from our splitting strategy which, unlike stratified methods, makes no attempt to maintain preserve the percentage of samples for each class. Our method is conceptually like “Leave One Group Out cross-validator” of a machine learning library (https://scikit-learn.org/1.5/modules/generated/sklearn.model_selection.LeaveOneGroupOut.html). The method splits data such that each training dataset consists of all samples except ones belonging to one specific group. In our case, plate information is provided as an array of integers that encodes the group of each sample. So, cell images from the same group (a plate or group of plates) stay together.

We have made two specific changes to improve precision:

1. We have revised the "Ensemble-based batch effect mitigation" section of Results to better explain our cross-validation approach

2. We have updated the terminology from "leave-one-plate-out cross-validation" to "leave-plates-out cross-validation" throughout the manuscript to more accurately reflect our methodology of using plate groups

Comment 16:

-Line 258: There is confusion because you suddenly introduce this binary variable R when you previously had a classification with multiple classes. It is not explained until the next section why this is the case, so you may want to include some of that background after you introduce R.

Response to Comment 16:

Thank you for pointing out this lack of clarity in our transition to binary classification. We have added the following explanation to the Discussion section (line 485) to clarify our rationale for using binary variables. “Our classification strategy employs distinct approaches for the model training and final compoun

---

## [Decision Letter · Decision Letter 1]

14 Feb 2025

Unveiling CNS Cell Morphology with Deep Learning: A Gateway to Anti-Inflammatory Compound Screening

PONE-D-24-44853R1

Dear Dr. Bae,

We’re pleased to inform you that your manuscript has been judged scientifically suitable for publication and will be formally accepted for publication once it meets all outstanding technical requirements.

Kind regards,

Carla Pegoraro

Staff Editor

PLOS ONE

Additional Editor Comments (optional):

Please note that Reviewer 4 had provided a rejection recommendation but that these comments do not need addressing. 

Reviewers' comments:

Reviewer's Responses to Questions

**Comments to the Author**

1. If the authors have adequately addressed your comments raised in a previous round of review and you feel that this manuscript is now acceptable for publication, you may indicate that here to bypass the “Comments to the Author” section, enter your conflict of interest statement in the “Confidential to Editor” section, and submit your "Accept" recommendation.

Reviewer #1: All comments have been addressed

Reviewer #3: All comments have been addressed

Reviewer #4: (No Response)

Reviewer #5: All comments have been addressed

2. Is the manuscript technically sound, and do the data support the conclusions?

Reviewer #1: Yes

Reviewer #3: Yes

Reviewer #4: No

Reviewer #5: Partly

3. Has the statistical analysis been performed appropriately and rigorously? 

Reviewer #1: Yes

Reviewer #3: Yes

Reviewer #4: No

Reviewer #5: N/A

4. Have the authors made all data underlying the findings in their manuscript fully available?

Reviewer #1: Yes

Reviewer #3: Yes

Reviewer #4: No

Reviewer #5: Yes

5. Is the manuscript presented in an intelligible fashion and written in standard English?

Reviewer #1: Yes

Reviewer #3: Yes

Reviewer #4: No

Reviewer #5: Yes

6. Review Comments to the Author

Reviewer #1: (No Response)

Reviewer #3: (No Response)

Reviewer #4: When a manuscript remains unsuitable after revisions, identify persistent issues in novelty, methodology, organization, and alignment with journal scope. Highlight specific sections requiring immediate improvement, such as experimental design, data analysis, or discussion depth. Provide clear, actionable suggestions to enhance clarity, originality, and relevance for publication readiness.

Reviewer #5: (No Response)

7. PLOS authors have the option to publish the peer review history of their article (what does this mean? ). If published, this will include your full peer review and any attached files.

**Do you want your identity to be public for this peer review?** For information about this choice, including consent withdrawal, please see our Privacy Policy .

Reviewer #1: **Yes: ** Earvin S. Tio

Reviewer #3: No

Reviewer #4: **Yes: ** Sachchida Nand Rai

Reviewer #5: No

---

## [Editor Report · Acceptance letter]

PONE-D-24-44853R1

PLOS ONE

Dear Dr. Bae,

I'm pleased to inform you that your manuscript has been deemed suitable for publication in PLOS ONE. Congratulations! Your manuscript is now being handed over to our production team.

Kind regards,

on behalf of

Dr Carla Pegoraro

Staff Editor

PLOS ONE